

# A Survey on High-energy Protons Response to Geomagnetic Storm in the Inner Radiation Belt

Zhaohai He[1], Jiyao Xu[1,2], Ilan Roth[3], Chi Wang[1], Lei Dai[1]

[1]State Key Laboratory of Space Weather, National Space Science Center, Chinese Academy of Sciences,

Beijing, China

[2]University of Chinese Academy of Sciences, Beijing, China

[3]Space Sciences Laboratory, University of California, Berkeley, California, USA

*Corresponding author: Zhaohai He (he_zh@nssc.ac.cn) and Jiyao Xu (xujy@nssc.ac.cn)*

**Abstract.** RBSPA observations suggest that the inner radiation belt high energy proton fluxes drop significantly during the storm main phase and recover in parallel to as the SYM-H index [Xu et al., 2019]. A natural problem arises: are these storm-time proton flux variations in response to the magnetic field modifications adiabatic? Based on Liouville's theorem and conservation of the first and third adiabatic invariants, the fully adiabatic effects of high energy protons in the inner radiation belt have been quantitatively evaluated. Two case studies show that theoretically calculated, adiabatic flux decreases are in good agreement with RBSPA observations. Statistical survey of 67 geomagnetic storms which occurred in 2013-2016 has been conducted. The results confirm that the fully adiabatic response constitutes the main contribution 90% to the changes in high energy protons in inner radiation belt during the storm main and recovery phases. It indicates that adiabatic invariants of the inner belt high energy protons are well preserved for majority of storms. Phase space density results also support adiabatic effect controls the varication of high energy protons especially for small and medium geomagnetic storms. Non-adiabatic effects could play important role for the most intense storms with fast changes in magnetic configuration.

## 1. Introduction

Previous experimental studies over many years in the terrestrial inner radiation belt were based on the

belief that the fluxes of high energy protons belt are fairly stable over years. However, later
measurements confirmed that significant changes in these fluxes have been observed in the outer
boundary of the inner radiation belt (1.7<L<2.5), in relation to large geomagnetic storms and solar
energetic particle events [e.g., Engel et al., 2015, 2016; Lorentzen et al., 2002]. Recently, we found that
sharp descent in high energy proton fluxes is accompanied by the corresponding depression of SYM-H

index even for the small geomagnetic storms (-50nT<Dst<-30nT) [Xu et al., 2019]. Therefore, in
contrast to previous belief the outer zone of the inner radiation belt is very sensitive to the geomagnetic
activities.

The most significant variations analyzed previously in the inner radiation belt included increase and loss

phenomenon on protons energies of MeVs, while in this paper we focus on proton loss events.
McIlwain [1966] reported that 40~110MeV proton Fluxes decrease during geomagnetic storm on April
17, 1965. Gussenhoven et al. [1994] and Hudson et al. [1997; 1998] showed high energy proton
depletions with CRRES observations. HEO-3 satellite observations show changes (~1–3 days) of 27 to

MeV protons in the outer zone of the inner belt (L ≥ 2.3) [Selesnick et al., 2010]. NOAA15,

NOAA16, and NOAA17 satellites observations show the responses of the inner radiation belt high-
energy protons to large geomagnetic storms (|Dst| > 200) [Zou et al., 2011]. Those studies focus mainly
on the timescale of more than tens of days, with some observations over timescale of months [Lorentzen
et al., 2002]. Recently, we found rapid response of high energy proton on the similar timescale of
geomagnetic activities in the inner radiation belt from RBSP satellite [Xu et al., 2019]. Even for the

weak disturbances, there is one-to-one correspondence between the proton fluxes descent and SYM-H
index depression.

Several mechanisms have been proposed to interpret high energy proton losses in the inner radiation
belt during storms. Dragt [1961] suggested that hydromagnetic waves of a few Hz could lead to high

energy proton losses. McIlwain [1965] suggested that non-adiabatic decreases of inner belt protons may



be associated with low-frequency magnetic fluctuations. Gussenhoven et al. [1994] proposed that proton depletions are caused by perturbation of the boundary of the inner zone. Anderson et al. [1997], Young et al.[2002] and Tu et al. [2014] suggested that proton are lost when their adiabatic motion breaks down due to magnetic field disturbances. Selesnick et al. [2010] and Zou et al. [2011] suggested
that the field line curvature scattering (FCS mechanism) is responsible for the drop of proton flux during intensive geomagnetic storms. Engel et al. [2015; 2016] found that the FCS mechanism is insufficient to support observed proton loss in the inner belt and they confirmed that the inductive electric field is very important and has a significant effect on the proton loss event.

The proton loss mechanisms mentioned above consist of non-adiabatic processes, while the fully adiabatic effect could also account for the flux dropout during magnetic storms. Those adiabatic flux changes known as the "Dst effect" have been applied mostly for relativistic electron flux decreases in the outer radiation belt [Li et al., 1997; Kim and Chan, 1997]. However, much less attention was devoted to adiabatic flux changes of high energy proton in the inner zone.

RBSP observations show that sharp descent in proton fluxes is accompanied by the corresponding depression of SYM-H index, with a one-to-one correspondence, regardless of the storm intensity [Xu et al., 2019]. Therefore it is natural to ask: are these storm-time proton flux variations fully adiabatic responses to magnetic field variations? How much of the main phase drop and recovery phase increase
are due to the adiabatic effects? The main purpose of this work is to evaluate quantitatively the extent to which fully adiabatic changes can account for the sharp descent and recovery in proton fluxes in the inner radiation belt observed during magnetic storms.

The organization of this paper is as follows. Section 2 describes the data and theoretical method to
evaluate the fully adiabatic effect used. Section 3 presents two case studies and a statistical survey on 67 geomagnetic storms during 2013-2016. Section 4 is summary and some general implications.





## 2. Data and methodology

### 2.1 Data accumulation

Van Allen Probes, including two spacecraft, RBSP A and RBSP B, are in orbits with perigee of about 600km and apogee of about 30,500km altitude. The satellites cover the regions of radiation belts in equatorial radial distance L from $L = 1.1$ to $L = 6.0$ and in geomagnetic latitude from $-20°$ to $20°$. The data of high-energy proton fluxes used in this paper are obtained from the Relativistic Electron-Proton Telescope (REPT) instrument on board RBSP A (Baker et al., 2012).

### 2.2 Quantitatively evaluated the fully adiabatic effects

The fully adiabatic effect has been widely used to account for the observed relativistic electrons variations in the storm time [Li et al., 1997; Kim and Chan, 1997]. It refers to the proton conservation of all the three adiabatic invariants, $\mu = p_\perp^2/2mB$, $J = \oint p_\parallel ds$, and magnetic flux $\Phi = \oint \vec{B} \cdot d\vec{S}$ as it experiences cyclotron, bounce and drift motions in magnetic field. Here $p_\perp$ and $p_\parallel$ represent the 90 perpendicular and parallel momentum components.

As the ring current develops during the storm time, the inner belt magnetic field decreases. To conserve the magnetic flux invariant, $\Phi$, the proton L shell increases. We assign the subscripts p, m and r to all the quantities at the prestorm time, storm main phase and recovery phase respectively. At prestorm time 95  $t_p$ protons drift in the $L_p$ shell with magnetic field strength $B_p$ and kinetic energy $E_p$. The corresponding prestorm time proton flux is denoted as $j_p=j (E_p, L_p; t_p)$. During the storm time, the inner belt magnetic field decreases from $B_p$ to $B_m$ and the proton L shell increases from $L_p$ to $L_m$. Additionally, for the conversation of the first invariant $\mu$, the proton energy decreases from $E_p$ to $E_m$. The storm time proton flux is denoted as $j_m=j (E_m, L_m; t_m)$. As the recovery phase approaches, the magnetic field configuration 100  always comes back to the approximate the prestorm time situation. The magnetic field intensity increases from $B_m$ to $B_r$; the proton energy increases from $E_m$ to $E_r$ and the proton L shell decreases from $L_m$ to $L_r$.

Liouville's theorem for trapped particles states that the phase space density $,f = j/p^2$ ,is constant along





the dynamical path of the particle [Roederer 1970]. It can be expressed by

$$f(\mu_p, J_p = 0, \Phi_p; t_p) = f(\mu_m, J_m = 0, \Phi_m; t_m) \qquad (1).$$

The relationship between the storm time kinetic energy $E_m$ and the prestorm time $E_p$ can be inferred from the constancy of the first adiabatic invariant,

$$\frac{p_p^2}{2mB_p(L_p)} = \frac{p_m^2}{2mB_m(L_m)} \qquad (2).$$

Here $p_p$ and $p_m$ are the proton momentums for prestorm time and storm time, respectively. The storm time proton flux $j_m$ is related to the prestorm time proton flux, $j_p$, as follows:

$$j(E_m, L_m; t_m) = \frac{B_m(L_m)}{B_p(L_p)} j(E_p, L_p; t_p) \qquad (3)$$

Thus, for adiabatic process the storm time proton flux is given in terms of the prestorm time proton flux multiplied by the magnetic field strength ratio, $B_m/B_p$.

**2.3 Magnetic Field Models**

Different magnetic field models have been adopted to calculate the fully adiabatic effect, such as Hilmer-Voigt symmetric ring current field model [Kim and Chan, 1997], modified dipole model [Selesnick and Kanekal, 2009; Tu and Li, 2011] and model developed by Tsyganenko et al [1989].

We present the predicted magnetic field from T89c (black curve) and modified dipole (blue curve) models and the observation data (red curve) from RBSPA satellite as shown in Figure 1. The SYM-H (Figure 1a and 1a') and Kp (Figure 1b and 1b') indexes have been shown as the input of magnetic field models. The results from two models during geomagnetic quiet (January 1, 2013) and active (March 17, 2013) periods have been compared. For L<3, the magnetic field from T89c and modified dipole models are almost coincident for both geomagnetic quiet and active times. It is indicated that T89c and modified dipole models have little differences in the inner radiation belt.

Therefore we chose the modified dipole model, which combines the Earth's dipole field $B_{dip}$ with a uniform southward magnetic field whose magnitude equals the Dst index. It can be expressed 130 by $B = B_{dip} + \delta B$. Here $B_{dip}$ is dipole field, $\delta B$ can be expressed by





$$\delta B = \begin{cases} -D_{st} \; for \quad D_{st} < 0 \\ \quad 0 \quad for \quad D_{st} > 0 \end{cases} \qquad (4).$$

It can be found in equation 8 of Selesnick and Kanekal [2009]. In this paper we use the SYM-H index
instead of Dst index.

## 2.4 Quiet Time Proton Flux Profile


To obtain a quiet time proton flux profile as a function of energy and L value, i.e., $j_p=j\,(E_p,\,L_p;\,t_p)$ in
equation (3), we used the differential proton flux data from the Relativistic Electron-Proton Telescope
(REPT) instrument on board RBSP A (Baker et al., 2012).

For the construction of an equatorially high energy flux profile, the data have been averaged during
relatively quiet periods over four years (2013-2016). We define the magnetic quiet time as AE index
less than 200nT and Kp index less than 2.

Figure 2(a) shows the average equatorial proton flux for three different intervals. Black curve represents

the average flux over one month in January 2013; blue curve represents the average flux over one
month in March 2015; red curve represents the average flux over four years from 2013 to 2016. Figure
2(b) shows the average proton fluxes for eight different energy channels at L=2.0 during those three
intervals.

As shown in Figure 2(a), the two month average fluxes vary much from the four year average data (red
line). This also can be seen from Figure 5 of Xu et al. [2019]' paper and Figure 2 in Selesnick et al.
[2016]'s paper, the high energy proton fluxes increase with time which may be caused by steady inward
diffusion. If we chose the four years average flux as initial input, the trend of fluxes increasing with
time has been neglected. That will cause the inaccuracy of the predicted data. Instead of the four year

average flux, we chose the month average data which do not exclude the four year trend of proton
fluxes. It is much more accurate for the quiet time average data over each month as input of prestorm
time flux than the average flux over four years period.





Based on the magnetic field model and quiet time proton flux profile, we evaluate proton adiabatic changes from prestorm time to storm time (including main phase and recovery phase). The calculating method is described as follows: (1) normalize the observed fluxes to geomagnetic equator location [Xu et al., 2019, equation 1 and 2]; (2) find $L_p$ and $E_p$ based on the first and third invariant with the known parameters $L_m=2.0$ and $E_m=21.25MeV$ in modified dipole magnetic field; (3) find j ($E_p$, $L_p$; $t_p$) based on the quiet time proton flux profile; (4) calculate j ($E_m$, $L_m$; $t_m$) from equation (3).

## 3. Observations

### 3.1 Case studies: two geomagnetic storms on 17 March 2015 and 20 January 2016

In order to investigate fully adiabatic effects on the high-energy protons during geomagnetic storms, two geomagnetic storms on 17 March 2015 and 20 January 2016 are chosen as examples.

The storm main phase begins when the first time SYM-H index falls below -15nT at the main phase and ends at the minimum SYM-H index. The recovery phase covers the times from the minimum of SYM-H index to the SYM-H index being recovered by 75%. The storm time intervals are highlighted by the three vertical blue dash lines as shown in Figure 3 and Figure 4.

Figure 3 shows the results of 17 March 2015 event, in which the left ($E_m=21.25$ MeV) and right ($E_m=27.6MeV$) columns refer to two different energy channel. The first and second rows in Figure 3 show the quiet time L shell $L_p$ calculated from the third invariant with the known parameters $L_m=2.0$ in modified dipole magnetic field. Energy $E_p$ has been calculated for $E_m=21.25MeV$ and $E_m=27.6MeV$ with the first invariant conservation respectively. Figure 3c and 3c' represent the quiet time proton flux, j ($E_p=21.25MeV$, $L_p$) and j ($E_p=27.6MeV$, $L_p$), for those two energy channels. Figure 3d and 3d' show the theoretically calculated fully adiabatic flux (the black line) and RBSPA observations (red dots) for two energy channels respectively. The bottom row in Figure 3 shows the geomagnetic activity SYM-H indexes. Figure 4 describes the geomagnetic storm of 20 January 2016, with the same format as Figure 3.





The observed flux data have been averaged for each orbit period during the interval. Both the calculated

and observed storm time proton fluxes vary with the same timescale of SYM-H index changes. We can

see that during the main phase the calculated fully adiabatic flux decreases is a fairly good agreement

with the observations from RBSPA satellite. During the recovery phase, the data slightly deviate from

calculated fully adiabatic fluxes, which indicate that there may be other non-adiabatic effects involved

during the recovery phase.

## 3.2 Statistical analysis: 67 geomagnetic storms during 2013-2016

Using the magnetic field models and the quiet time proton flux profile from the RBSP observations as

inputs, we can calculate the fully adiabatic storm time proton flux variations for a given local magnetic

field intensity during 67 geomagnetic storms as mentioned at Table 1 in Xu et al. [2019]'s paper. Figure

5 shows the statistical results for the storm main phase (Fig.5a) and recovery phase (Fig.5b).

Figure 5 shows the result of predicted and observed proton fluxes for $E_m=21.25$MeV. As shown in

figure 5, the abscissa is the data and the ordinate is theoretically calculated flux with equation (3). The

red line is the fit line y=ax (a=0.913 for main phase and 0.929 for recovery phase). The correlation

coefficient between the predicted and observed fluxes is 0.89 and 0.88 for two phases respectively. As

we expected, the predicted fluxes are highly correlated with the observed fluxes for both phases.

Therefore the fully adiabatic effect may primarily contribute for high energy protons decrease and

recovery in inner radiation belt during storm main phase and recovery phase.

Additionally, from the figure 5, it can be seen that there exist differences between the observed fluxes

and the predicted fully adiabatic fluxes. The fully adiabatic effect contributes about 90% of proton

descent and recovery, which means that some non-adiabatic loss mechanisms exist, such as

low-frequency magnetic fluctuations and field line curvature scattering as mentioned above [McIlwain,

1965; Anderson et al., 1997; Young et al.,2002; Tu et al.,2014; Engel et al., 2015; 2016]. Therefore, the

non-adiabatic effects could also play important roles for the recovery of high-energy protons in the

storm recovery phase and could not be neglected.

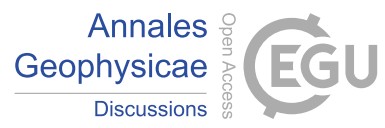

### 3.3 Analysis on phase space densities during 2013-2016

The proton phase space density can be deduced from Equation (1) in Chen et al. [2005] with the observed flux data and modified dipole field. As shown in Figure 6, (a), (b), and (c) are SYM-H index and two channels of proton fluxes (21.25MeV and 27.6 MeV), which is same from Figure 3 in Xu et al. [2019]. We represent them for convenience as they compare with the results of phase space density. Figure (d) and (e) are phase space density for u=535MeV/G and u=700MeV/G. the vertical dash lines indicates 67 geomagnetic storms during 2013-2016.

It can be seen from Figure 6 (d) and (e) that for small and modest geomagnetic storms the phase space density varies less before and after storms. It is indicated that phase space density maintains constant and it support our previous results form case studies and statistical survey. For large storms it can be seen that phase space density varies much before and after storms. It means that the nonadiabatic process may be involved during the intense geomagnetic activities. The mechanisms deplete the high energy protons should be analyzed for further studies.

### 4. Summary

In this paper, we found that the high energy proton fluxes at the inner radiation belt decrease significantly during the storm main phase and recover as the SYM-H index recovers. It seems that the variations of fluxes have similar timescale to the changes of local magnetic fields and SYM-H index. Based on those observations, Liouville's theorem and the conservation of the first and third adiabatic invariants have been used to test the fully adiabatic effects of high energy protons in the inner radiation belt during the storm main and recovery phase. Both case studies and statistical surveys have been conducted to quantitatively evaluate the adiabatic effects.

From the studies on 17 March 2015 and 20 January 2016, we find that the calculated fully adiabatic flux decreases is in fairly good agreement with the data from RBSPA satellite during the main phase and recovery phase. Both the calculated and observed storm time proton fluxes vary with the same timescale of SYM-H index changes.





We repeatedly calculated the fluxes from the prestorm data as input for 67 geomagnetic storms which occurred during 2013-2016. The correlation coefficients are 0.89 and 0.88 for storm time and recovery phase respectively. These high-correlation coefficients indicate that the fully adiabatic effect does primarily contribute to the inner radiation belt high energy protons decrease during storm main phase and increase during recovery phase. Phase space density always maintains constant before and after

small and medium geomagnetic storms. It support our results form case studies and statistical survey. It can also be seen that phase space density varies much before and after storms for large and huge storms. It means that the nonadiabatic process may be involved during those intense geomagnetic activities. Therefore the mechanisms for depletion of the high energy protons during very intense activities should be analyzed for further studies.


The fully adiabatic effect contributes about 90% of proton flux descent and recovery. Since adiabatic modifications form the main contribution of protons in the inner radiation belt due to geomagnetic storms, statistically the recovery phase of the belt results in final state very similar to the initial one. This should be contrasted with the electron fluxes at the outer radiation belt where statistical study

found that only in 28% of storms the final state was not distinguishable from the initial, in 53% the final flux was significantly higher, while in 19% was lower [Reeves et al., 2003]. The reason for the inner belt quasi-adiabatic behavior is due to the stronger magnetic field at lower L which requires more intense external perturbation as well as slower response of the high-mass protons (in comparison to electrons) to these oscillations. Still, the nonadiabatic processes require additional investigation since in

the more intense geomagnetic storms with stronger temporal changes they become crucial for understanding the energetic proton behavior in the inner radiation belt.

**Acknowledgments**

This work was supported by the National Natural Science Foundation of China (41831073 and 41674152), by the Strategic Priority Program on Space Science, Chinese Academy of Sciences, grants

XDA15052500 and XDA15350201, and by the Open Research Project of Large Research Infrastructures of CAS―"Study on the interaction between low/mid-latitude atmosphere and ionosphere





based on the Chinese Meridian Project". I. R. acknowledges the support of NASA grant NNN06AA01C.

We thank the Van Allen Probes REPT and EMFISIS teams, and the data were downloaded online (ftp://cdaweb.gsfc.nasa.gov/pub/data/rbsp/). We acknowledge the World Data Center for

Geomagnetism, Kyoto, for the provisional SYM-H indices data (http://wdc.kugi.kyoto-u.ac.jp/).

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

**Figure Caption**

Figure 1. Magnetic field calculated from T89c and Modified dipole models and observation data from RBSPA satellite during quiet (Jan 01 2013) and active (March 17 2015) times. Figure (a) and (a') Kp index, figure (b) and (b') SYM-H index, and Figure (c) and (c') magnetic field. Dashed lines indicate L shell at 3.0.

Figure 2. (a) Magnetic quiet time (AE<200nT and Kp<2) proton flux data in the inner radiation belt for three different time intervals (Jan. 2013, Mar. 2015 and four years from 2013-2016). (b) Proton flux for eight different channels during those three intervals.

Figure 3. Calculation results for the adiabatic flux variations ($j_m$) at $L_m$=2.0 for high energy protons with energy $E_m$=21.25MeV (left) and Em=27.6MeV (right) on 17 Mar. 2015 geomagnetic storm. Figures 3a−3e and 3f−3j give intermediate results for the $j_m$ calculation, showing (a, f) quiet time L shell $L_p$, (b, g) energy Ep for $E_m$=21.25MeV and Em=27.6MeV, (c,h) quiet time proton flux profile $J_p$, (d,i) the resulting $J_m$, (e,j) the SYM-H index.

Figure 4. Calculation results for the adiabatic flux variations (jm) on 20 Jan. 2016 geomagnetic storm. The same format with Figure 3.



Figure 5. Statistical survey for calculated data based on Liouville's theorem and first invariant for two different channels (upper for 21.25MeV and bottom for 27.6MeV) and phases (Left for main phases and right for recovery phases).

Figure 6. Profiles for SYM-H index (a), flux from RBSP (b and c), and phase space density (d and e) during 2013-2016 interval. Vertical dash lines indicate the geomagnetic storms occurred in four year interval.





Figure 1

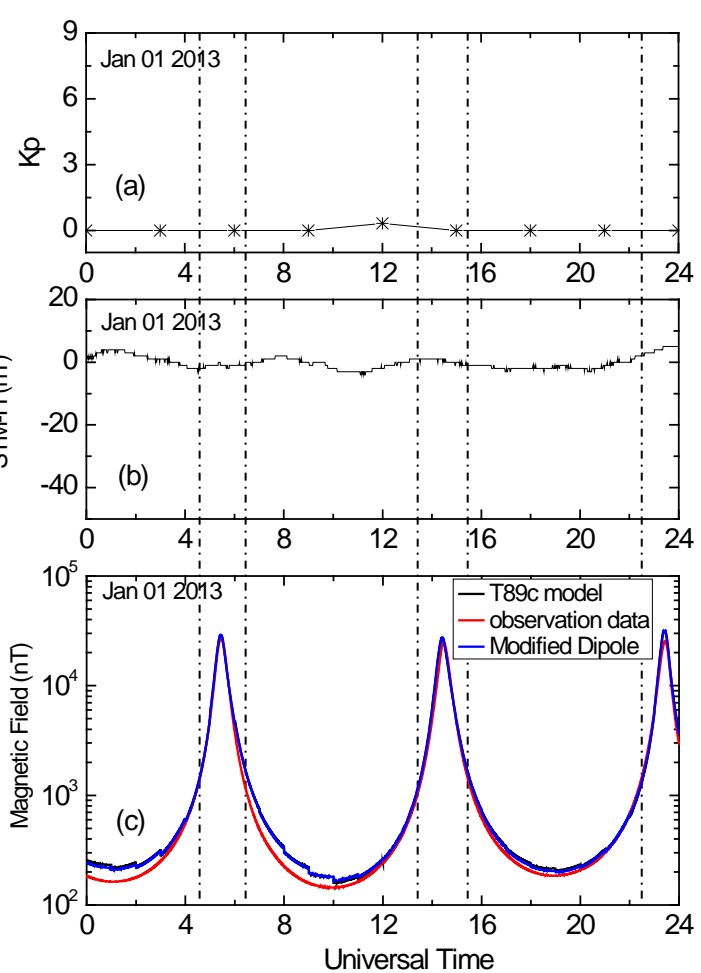

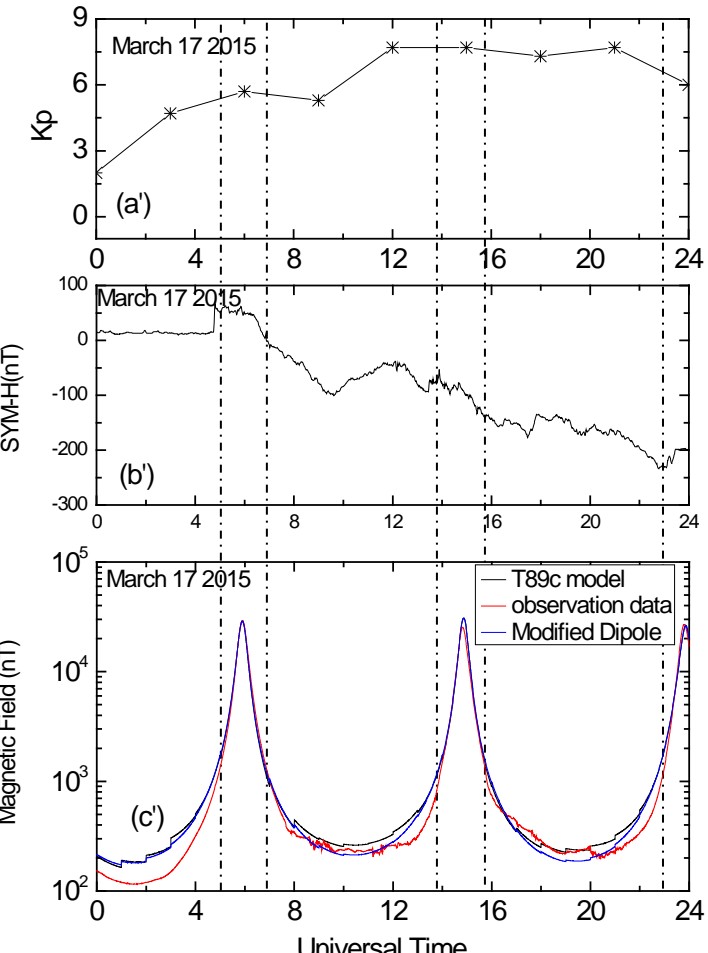





354    Figure 2
355

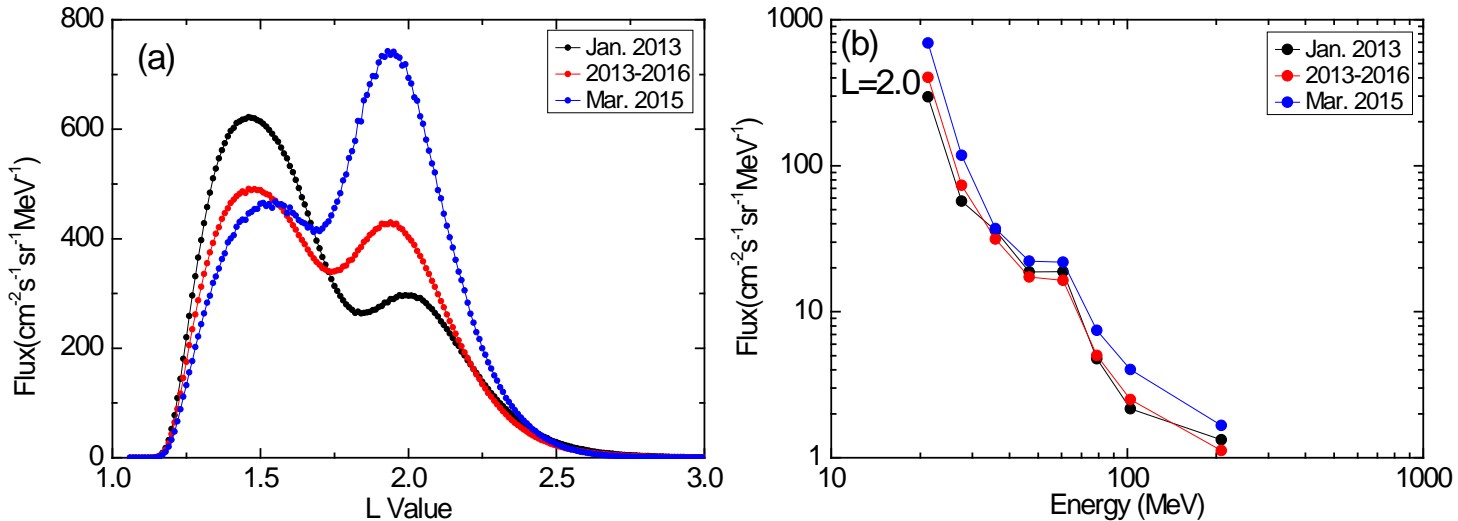



356    Figure3





357     Figure 4



358      Figure 5





359    Figure 6



360





361



362

