# Peer review of "A Survey on High-energy Protons Response to Geomagnetic Storm in the Inner Radiation Belt"

_Annales Geophysicae, 2021_

## Author Comment (AC1)

Reply to Referees

Firstly, we thank for reviewers' comments and questions. They are beneficial to our research works. The point-to-point responses are as follows:

Reply to **Comment on angeo-2021-4**

Anonymous Referee #1

This paper claims to show that adiabatic effects dominate the changes in proton fluxes in the outer part of the proton radiation belts (sometimes referred to as the inner zone).

A first (more minor point compared to my latter concerns) is that much of the cited literature concerns changes in the proton belts that are more long-lasting. For example, losses due to field line curvature scattering are true losses as opposed to temporary (adiabatic) changes. This paper does not contradict those other studies. It has a different objective.

The major problems with this paper are that (a) the methodology is not presented with enough detail to understand how the authors actually analyzed the data and (b) the methodology itself appears responsible for the results that are presented.

Specifically, the authors present formulas that they claim quantify the changes in flux due to adiabatic processes that preserve mu and L. Those are listed in equations 1-4 which represent the flux during the storm (subscript m) as a function of flux prior to

the storm (subscript p). The two are related by three variables: Energy, L-shell, and

Magnetic field strength. The variables during and prior to the storm are represented by

_p and _m.

The first problem is that the equations that relate E_m to E_p and L_m to L_p are not

given so the quantities in figure 3 cannot be verified.

We have added the equation (3) and (4), from which we can infer the parameters E_p

and L_p. They are obtained from the conversation of fist and third invariants. In the

paper they are emphasis with red color at line 120-127.

The second problem is that figure 3 plots E_p, L_p, and j_p as a function of time for

fixed values of L_m and E_m. Surely it should be the other way around. For a given

pre-storm condition (_p) the quantities during the storm (_m) are a function of time. It

is not at all helpful to present it in terms of the "pre-storm" conditions vary as a

function of time during the storm.

Let us explain the "pre-storm" conditions vary as a function of time during the storm.

Firstly, as we know parameters Lm=2.0 and Em=21.25MeV, then we can infer

parameters Lp and Ep based on the first and third invariant [equation 3 and 4 in our

paper, equation 3 and 4 are added lately] and magnetic field model. Secondly, we

constructed the quiet time flux profile over four years. We obtained monthly average

fluxes with L bin from 1.1 to 3.0 and width 0.01 for all 48 months like black and blue

lines in Figure 2(a). Each energy channel for each month has been analyzed

repeatedly like the black and blue lines in Figure 2(b). Therefore we got the quiet time flux distribution J (E, L), which actually is the function of parameters E and L. During the main phase, the magnetic field changes and the corresponding Ep and Lp are also changes. Thirdly, we have obtained Lp and Ep at above obtained, and we can trace back the quiet time J (Ep, Lp) by linear interpolation with parameters Lp and Ep. We added some information at line 164-166 and 188-190.

The third, and biggest, problem is that the relationship between all of the variables (e.g.L_p to L_m, E_p to E_M) are all a function of B_p/B_m. Since B == B_dip +dB and dB =-symH (for symH<0) then all of the pre-storm and storm-time variables are related to one another as a function of dB == -symH. This can be seen very clearly in figure 3 where all predicted variables follow every bump and wiggle of symH.

The similarity between the variation in SYM-H index and the all predicted variables appear naturally for fully adiabatic flux changes. The prestorm time Lp and Ep changes roughly linear with decreasing SYM-H index, so from equation (5) and the quiet time proton flux model, the storm time proton flux Jm decreases exponentially with SYM-H index; thus the changes in the parameter Jm will be similar to changes in SYM-H index.

For true calculations of adiabatic effects the radial gradients of PSD are critical (as is the second invariant which is ignored here). For example, a flat radial gradient produces no change in flux when B changes. This analysis simply samples the fluxes

(j_p) at different values of L and E that are related to an arbitrarily-chosen value of L_m and E_m where the relationship is defined by symH. It is a totology to conclude that adiabatic changes (defined by dB == -symH) "explain" the flux variations.

We have revised Figure 6. In Figure 6 we present two PSD radial distributions in time inferred from different L*. We also have computed the ratio for two different L* and we find the ratio for storm time is comparable to the quiet time. In Figure 6 (d) and (e), PSD changes not much before and after storm time in most small and medium intensive cases. We agree that the flat radial gradient produces no changes in fluxes when magnetic field changes. We don't know if we have answered your question accurately. If not, please make comments.

The brief discussion of phase space density in section 3.3 does not contain enough information to know what the authors have done or what is being plotted in figure 6. Is the PSD at fixed third invariant (L*)? If so, what L*? It is currently impossibe to know if the PSD results support the preceding conclusions or not.

The proton phase space density can be deduced by equation $f_{ch} = \left\{ \frac{j_{ch}}{\langle p^2 c^2 \rangle_{ch}} [1.66 * 10^{-10}] \right\} * 200.3$ (Equation (1) in Chen et al. [2005]) with the observed flux data and modified dipole field. jch is the flux and $\langle p^2 c^2 \rangle_{ch} = 0.5 * [K_{min}^{ch}(K_{min}^{ch} + 2m_0 c^2) + K_{max}^{ch}(K_{max}^{ch} + 2m_0 c^2)]$. $K_{min}^{ch}$ and $K_{max}^{ch}$ are the lower and upper limit of each energy channel in MeV respectively. $m_0 c^2$ is the rest energy of an proton. The L* is defined as Roederer [1970], $L^* = 2\pi k_0/(\Phi R_E)$, where k0 is the earth's dipole magnetic moment, RE is the earth's radius, and $\Phi$ is the third invariant.

Figure (d) and (e) are phase space density for u=535MeV/G and u=700MeV/G at equatorial plane (J=0) for two different L* (black represents L*=2.0 and red for L*=2.3).

We added some information at line 247-253 and 256-258.

---

## Author Comment (AC2)

Reply to Referees

Firstly, we thank for reviewers' comments and questions. They are beneficial to our research works. The point-to-point responses are as follows:

**Comment on angeo-2021-4**

Anonymous Referee #2

Referee comment on "A Survey on High-energy Protons Response to Geomagnetic Storm in the Inner Radiation Belt" by Zhaohai He et al., Ann. Geophys. Discuss., https://doi.org/10.5194/angeo-2021-4-RC2, 2021

I apologize to the authors and the editor for waiting until the very last minute to submit my review, especially since the other reviewer was so prompt.

Substantive comments:

Line 35: Should "proton loss events" be "proton loss events at [specific energy range]", in contrast to the "energies of MeVs" previously studied?

In our paper we found that high energy protons with fist three channels (18.5MeV~40.6MeV) of REPT instrument on RBSP-A change with geomagnetic activities. We add the specific energy range in our manuscript with red colors at line 35-36.

Section 2.1 on the data is extremely short. Are the data to be used selected in any way (pitch angle, closeness to the equator, …), or are all data in a given time period just averaged together? What energy ranges will be used?

The proton data are spin-averaged differential proton flux, which are same with our previous work [Xu et al., 2019]. Three channels (18.5~24.0MeV, 24.0~31.2MeV and 31.2~40.6MeV) have been investigated and here we present first two channels in detail. These proton fluxes have been projected to the equator with normalization equations (Equation 1 and 2 in Xu et al. [2019]).

The magnetic field data are from the Electric and Magnetic Field Instrument and Integrated Science (EMFISIS) [Kletzing et al., 2013] instrument and their resolution is 1 second. Geocentric solar magnetospheric (GSM) coordinate system has been adopted for comparing the observed and model predicted magnetic field in this paper.

As suggested by the reviewer, we added this information of science data in our paper at line 84-94.

Line 87: probably "proton conservation" -> "particle's conservation" generally, as electrons are referenced in the preceding sentence

We corrected the expression in our paper at line 96.

Line 93: The authors should make clear the particular definition of L (L*, McIlwain L, simple geometric result for drift orbit radius divided by Earth radius) by which the data will be sorted and labeled.

In Section 2.2, we define the L value as a radial distance of the drift shell, which has

been added at line 102-103.

Line 96: "During the storm time" -> "During the storm main phase"; likewise on line 98

We corrected them at line 106 and 108.

Equation 1: This is the only time J = 0 is mentioned; are only data that have B/Beq close to unity kept for analysis, or are all data around each orbit simply considered to be equatorial?

Here J=0 means only equatorial data with B/Beq less than 1.05 have been analyzed in our work. The information has been added at line 116.

Line 118: Should mention T89c here to link it with the Tsyganenko et al. (1989) reference; most readers will know that's what is meant, but not all.

We added T89c magnetic field model in our paper at line 135.

Line 120: perhaps "predicted magnetic field along the RBSP-A orbit from T89c …"

We rewrote the sentence at line 138-140.

Predicted magnetic field along the RBSP-A orbit from T89c (black curve) and modified dipole (blue curve) models and the observation data (red curve) from EMFISIS instrument on RBSPA satellite have been presented in Figure 1.

Line 121: It would be good to identify which instrument aboard RBSP-A provided the measured magnetic field magnitude.

The magnetic field data are from the Electric and Magnetic Field Instrument and Integrated Science (EMFISIS) [Kletzing et al., 2013] instrument and their resolution is 1 second. Geocentric solar magnetospheric (GSM) coordinate system has been adopted for comparing the observed and model predicted magnetic field in this paper. We added the magnetic field instrument information in the paper at line 90-93 and 138-140.

Lines 133-134: Why use SYM-H instead of Dst, if Dst is the parameter used in the definition of the modified dipole model (equation 4)?

As mentioned in equation 4 and Selesnick and Kanekal [2009], Dst is used in the definition of the modified dipole model. Usually the Dst have a resolution of one hour, while the theoretical calculating in our paper needs higher resolution of Dst. As we know the SYM-H index is very similar to the Dst index and it has a resolution of one minute. Therefore we use the SYM-H index instead of Dst index. we added some information at line 151-153.

Lines 152-153: I'm not sure of the time intervals reported by the two references, but might increases in proton flux be due to solar modulation of the CRAND sources and losses rather than "steady inward diffusion"?

Sorry, we did not express well in the sentences. Selesnick et al [2016] shows steady

increase of monthly trapped proton intensity during October 2013 to August 2015 and they postulated that these changing intensity of protons were consistent with inward diffusion of trapped solar protons. In Xu et al [2019], we normalized all the data to the equator and the bump in 2015 is much more obvious, while we suggest that there is a more efficient mechanism than the inner ward diffusion. We agree with reviewer's opinion, which the increases in proton flux might be due to solar modulation of the CRAND sources and losses rather than "steady inward diffusion". We rewrite the sentence at line 174-175

Figure 2(a): What energy is shown in this L profile?

Figure 2(a) shows the first energy channel E=21.25MeV. We revised it in our text at line 166-167.

Lines 155-156: The meaning of "the month average data which do not exclude the four year trend of proton fluxes" is not clear to me.

There is an increasing trend of observation data from 2013 to 2016. If we use the four year average data, which cannot reflect the actual data trend, for year 2013 and 2014 the initial inputs are larger than observation data and for year 2015 and 2016 the initial inputs are less than observation data. That will cause the inaccuracy of the predicted output of fluxes. For monthly average data, the increasing trend still exists and it is closer to the real initial input. Our expression is not so accuracy and we have revised them at line 177-181.

Lines 161-162: Ah – here we find the relationship between the data and the magnetic equator. Perhaps back in section 2.1 it would be good to say that "proton fluxes are projected to the equator based on the pitch-angle distributions fitted by Xu et al. (2019)" or something.

These proton fluxes have been projected to the equator with normalization equations (Equation 1 and 2 in Xu et al. [2019]).

We added this information at line 87-88.

Lines 163-164: It is not clear how to "find j(Ep, Lp;tp) based on the quiet time flux profile". Figure 2(a) gives the L dependence at an unlabeled energy, and figure 2(b) gives the E dependence at L=2, in all cases with values differing by a factor of two in places for three different times. Which time is selected? Are the profiles assumed to be separable (that is, j(E,L) = A(E) * B(L) with A & B given by the panels of figure 2), or is there a database from which these are sampled?

At first, as we know parameters Lm=2.0 and Em=21.25, then we can find Lp and Ep based on the first and third invariant [equation 3 and 4 in our paper, equation 3 and 4 are added lately] and magnetic field model. Second, we constructed the quiet time flux profile over four years. We obtained monthly average fluxes with L bin from 1.1 to 3.0 and width 0.01 for all 48 months like black and blue lines in Figure 2(a). The energy shown in Figure 2(a) is 21.25MeV. Each energy channel for each month has been analyzed repeatedly like the black and blue lines in Figure 2(b). Therefore we

got the quiet time flux distribution J (E, L). Third, we have obtained Lp and Ep at above obtained, and we can trace back the corresponding J (Ep, Lp) by linear interpolation of Lp and Ep. We added some information at line 164-166 and 186-188.

Line 172: By "recovered by 75%," do you mean 75% of the way back to zero, or to -15nT?

We did not express completely. Here recovered by 75% means 75% of the way back to zero. We revised it at line 198.

Figure 3: Panel labels "(a) Lp" etc. are very small; also, the labels in the figure and the caption are a-j, whereas in the main text they are referred to as a and a', b and b', etc.

We enlarge the labels. The captions a-j have been replaced as a-e and a'-e', so we corrected at line 381-383.

Lines 179-180: The sentence describing panels 3c and 3c' appears to refer to fixed values of Ep = 21.25 MeV and 27.6 MeV, but the previous sentence refers to calculation of a time varying Ep corresponding to each of two fixed values of Em. Since the observations in panels 3d and 3d' are presumably each for a fixed energy Em, does that mean that panels 3c and 3c' are fluxes for time-varying Ep and Lp?

Sorry, we made mistakes. Figure 3c and 3c' is quiet time fluxes profile J(Ep, Lp), in which Ep and Lp are presented at Figure 3a, 3b, 3a' and 3b'. The Ep and Lp are time

varying parameters and Figure 3c and 3c' are quiet time fluxes for time-varying Ep and Lp. We rewrite the sentence at line 206-208.

Line 189: The data "slightly deviate" from the calculated fluxes; but the red points jump up and down from one to the next by an amount that exceeds the difference between the red dots and black curves. Might this be due to orbit/attitude interactions that are incompletely corrected for when data are projected to the magnetic equator? Can you estimate the magnitude of this error?

We estimate the magnitude of error between the observed and predicted data with root-mean-square prediction error (RMSPE),

$$RMSPE = \frac{1}{n}\sqrt{\sum_{i=1}^{n}(\frac{y_{obs} - y_{pre}}{y_{obs}})^2}$$

Here $y_{obs}$ and $y_{pre}$ represent observation and predicted data. It has been calculated for two energy channels and two different phases (storm main and recovery phases). They are all less than 1% (0.52% and 0.62% for two energy channels in main phase; 0.82% and 0.9% for two energy channels during recovery phase.) Here the calculated fluxes are all equatorial data, and we think the orbit/attitude interaction is not the main effect for the "slightly deviate". The most effect factors of "slightly deviate" might be the initial equatorial proton fluxes distribution. We added some information of RMSPE at line 217-225.

Line 208: As per line 189, do the systematic errors in the data allow the conclusion

that "some non-adiabatic loss mechanisms [must] exist"?

We think the fully adiabatic effect contributes about 90%, and this is from the fit results of observation and prediction data y=ax. We get y=0.912x for main phase and y=0.929x for recovery phase. And the RMSPE results are all less than 1%, we think there might be some non-adiabatic loss mechanism exist. "Not must" we have revised at the line 240-242.

Figure 6: Panels d and e are labeled with both an energy and a mu/J combination. The energy of a fixed mu and J will vary with magnetic conditions; are the phase space densities in each panel calculated from the flux measured at that time-varying energy (with the label giving the energy of that mu/J channel during quiet times), or are they simply the fluxes at the constant labeled energy scaled to a phase space density?

We have revised Figure 6. We added some information about what we have done on PSD.

The proton phase space density can be deduced by equation $f_{ch} = \left\{\frac{j_{ch}}{\langle p^2 c^2 \rangle_{ch}}[1.66 * 10^{-10}]\right\} * 200.3$ (Equation (1) in Chen et al. [2005]) with the observed flux data and modified dipole field. jch is the flux and $\langle p^2 c^2 \rangle_{ch} = 0.5 * [K_{min}^{ch}(K_{min}^{ch} + 2m_0 c^2) + K_{max}^{ch}(K_{max}^{ch} + 2m_0 c^2)]$. $K_{min}^{ch}$ and $K_{max}^{ch}$ are the lower and upper limit of each energy channel in MeV respectively. $m_0 c^2$ is the rest energy of an proton. The L* is defined as Roederer [1970], $L^* = 2\pi k_0/(\Phi R_E)$, where k0 is the earth's dipole magnetic moment, RE is the earth's radius, and $\Phi$ is the third invariant.

Figure (d) and (e) are phase space density for u=535MeV/G and u=700MeV/G at equatorial plane (J=0) for two different L* (black represents L*=2.0 and red for L*=2.3).

We added some information at line 247-253 and 256-258.

Typographical suggestions:

In general, the paper would benefit from a thorough proofreading for language. A few instances that I noted in passing:

Lines 19-20: "support adiabatic effects controls the varication" -> "support adiabatic effects controlling the variation"

Line 56: intensive -> intense

Line 98: conversation -> conservation

Line 110: momentums -> momenta

Line 212: "could not be" -> "should not be"

Line 242: "high-correlation" -> "high correlation"

Line 244: maintains -> remains

Line 245 "support our results form" -> "supports our results from"

We have corrected all these mistakes in our paper.